# Diversity and evolution of computationally predicted T cell epitopes against human respiratory syncytial virus

Jiani Chen[1,2,3,4], Swan Tan[1,3,4,5], Vasanthi Avadhanula[6], Leonard Moise[3,7¤], Pedro A. Piedra[6], Anne S. De Groot[3,7], Justin Bahl[1,2,3,4,5,8] *

**1** Center for Ecology of Infectious Diseases, University of Georgia, Athens, Georgia, United States of America, **2** Institute of Bioinformatics, University of Georgia, Athens, Georgia, United States of America, **3** Center for Vaccines and Immunology, University of Georgia, Athens, Georgia, United States of America, **4** Center for Influenza Disease and Emergence Response, University of Georgia, Athens, Georgia, United States of America, **5** Department of Infectious Diseases, University of Georgia, Athens, Georgia, United States of America, **6** Department of Molecular Virology and Microbiology, Baylor College of Medicine, Houston, Texas, United States of America, **7** EpiVax Inc., Providence, Rhode Island, United States of America, **8** Department of Epidemiology and Biostatistics, University of Georgia, Athens, Georgia, United States of America

¤ Current address: Seromyx Systems Inc., Woburn, Massachusetts, United States of America
* justin.bahl@uga.edu

**Data Availability Statement:** Data and materials availability: Sequence data was retrieved from NCBI's GenBank nucleotide database (https://www.

## Abstract

Human respiratory syncytial virus (RSV) is a major cause of lower respiratory infection. Despite more than 60 years of research, there is no licensed vaccine. While B cell response is a major focus for vaccine design, the T cell epitope profile of RSV is also important for vaccine development. Here, we computationally predicted putative T cell epitopes in the Fusion protein (F) and Glycoprotein (G) of RSV wild circulating strains by predicting Major Histocompatibility Complex (MHC) class I and class II binding affinity. We limited our inferences to conserved epitopes in both F and G proteins that have been experimentally validated. We applied multidimensional scaling (MDS) to construct T cell epitope landscapes to investigate the diversity and evolution of T cell profiles across different RSV strains. We find the RSV strains are clustered into three RSV-A groups and two RSV-B groups on this T epitope landscape. These clusters represent divergent RSV strains with potentially different immunogenic profiles. In addition, our results show a greater proportion of F protein T cell epitope content conservation among recent epidemic strains, whereas the G protein T cell epitope content was decreased. Importantly, our results suggest that RSV-A and RSV-B have different patterns of epitope drift and replacement and that RSV-B vaccines may need more frequent updates. Our study provides a novel framework to study RSV T cell epitope evolution. Understanding the patterns of T cell epitope conservation and change may be valuable for vaccine design and assessment.

## Author summary

Lower respiratory infections caused by human respiratory syncytial virus (RSV) is a global health challenge. B cell epitope immune response has been the major focus of RSV vaccine

ncbi.nlm.nih.gov/nucleotide/). Accession number to RSV sequences used in this study are available in supplementary materials. Code to generate T epitope landscapes are deposited in GitHub https://github.com/JianiC/RSV_Epitope.

**Funding:** This work was supported by the National Institute of Allergy and Infectious Diseases, a component of the NIH, Department of Health and Human Services, under contract 75N93019C00052 (J.C. and J.B.) and Contract No. 75N93021C00018 (NIAID Centers of Excellence for Influenza Research and Response, CEIRR) (J.B.). J.B and S. T. are funded in part from the Centers for Disease Control under Contract No. 75D30119C06826 and 75D30121C11990. The funders had no role in study design, data collection and analysis, decision to publish, or preparation of the manuscript.

**Competing interests:** We have read the journal's policy and the authors of this manuscript have the following competing interests: ADG is a senior officer and majority shareholder of EpiVax, Inc. Some of the epitope prediction tools used in this study were developed by EpiVax. The tools were available for academic collaboration at no cost. LM was employed by EpiVax while this study was conducted.

and therapeutic development. However, T cell epitope induced immunity plays an important role in the resolution of RSV infection. While RSV genetic diversity has been widely reported, few studies focus on RSV T cell epitope diversity, which can influence vaccine effectiveness. Here, we use computationally predicted T cell epitope profiles of circulating strains to characterize the diversity and evolution of the T cell epitope of RSV A and B. We systematically evaluate the T cell epitope profile of RSV F and G proteins. We provide a T cell epitope landscape visualization that shows co-circulation of three RSV-A groups and two RSV-B groups, suggesting potentially distinct T cell immunity. Furthermore, our study shows different levels of F and G protein T cell epitope content conservation, which may be important to correlate with duration of vaccine protection. This study provides a novel framework to study RSV T cell epitope evolution, infer RSV T cell immunity at population levels and monitor RSV vaccine effectiveness.

## Introduction

Human respiratory syncytial virus (RSV) is a negative-strand RNA virus that is classified in the *Orthopneumovirus* genus of the family *Pneumoviridae*. It is a major cause of lower respiratory disease in young infants, immunocompromised individuals, and elderly people, resulting in annual epidemics worldwide [1]. The single-stranded RNA genome of RSV is approximate 15.2 kb and encodes 11 viral proteins [2]. The Fusion (F) and Glycoprotein (G) proteins are the two major surface proteins [3]. F protein is generally thought to be conserved and therefore it is the focus of most current RSV vaccine designs. Although G protein is highly variable, its contribution to disease pathogenesis and its role in the biology of infection suggest it can also be an effective RSV vaccine antigen [4]. Despite the significant burden of RSV infection worldwide, there is no licensed vaccine. The only approved intervention is passive immuno-prophylaxis with palivizumab, which is achieved by administering the monoclonal antibody (mAb) to a highly restricted group of infants under the age of 24 months and treatment must be repeated monthly during the RSV season due to the relatively short half-life of the antibody [5,6]. Due to the high cost of monoclonal antibody treatments, this intervention is limited to high-risk infants and is generally unavailable in developing countries. An RSV vaccine is an urgent global healthcare priority, and it is likely that different strategies are needed for the various high-risk groups.

A number of research teams have worked on the development of RSV vaccine since its isolation and characterization in 1956 [7,8]. However, vaccination with the formalin-inactivated, alum precipitated RSV (FI-RSV) vaccine in RSV-naïve infants and young children, led to the development of vaccine enhanced disease (VED) that hampered vaccine development for decades to follow [9]. Many studies have been conducted to explain this undesirable outcome. It is likely that formalin fixation led to a vaccine that mostly presented the post-fusion conformation of RSV F protein, leading to an excess of non-neutralizing antibodies and immune complex formation [10–12]. Other studies indicated that an impaired T cell response with Th2 skewing [13,14], as well as complement deposition in the lungs, contributed to enhanced neutrophil recruitment [12]. Recent developments, including the resolution of the F protein [15] and the development of RSV rodent models [16] have contributed to a number of vaccine candidates with novel designs and formulations currently in clinical trials [3,17,18].

While most current RSV vaccination strategies focus on a B-cell-induced neutralization immune response, T cell immunity also plays a major role in the resolution of virus infection and is essential for RSV vaccine development [17,18]. Once RSV infection of the lower airways

is established, CD8 T cells play an important part in viral clearance and CD4 helper T cells can orchestrate cellular immune responses and stimulate B cells to produce antibodies. However, Th2-biased responses have been associated with animal models of RSV VED, and measurement of Th1 and Th2 responses are considered important to predict the safety of vaccine candidates [12]. Therefore, induction of a balanced cell-mediated immune response through vaccination would promote RSV clearance, but caution must be taken to avoid the potential for immunopathology. Taken together, a closer examination of T cell immunity and the virus sequences that induce T cell responses are needed for RSV vaccine development.

Human respiratory syncytial virus has a complex circulation pattern in the human population. Within two antigenic groups, RSV-A and RSV-B, different genotypes can co-circulate within the same community, while novel RSV genotypes with high genomic diversity may arise and potentially replace the previously dominant genotypes [19]. In recent years, several unique genetic modifications in RSV have been identified, including a 72-nucleotide (nt) duplication (ON genotype) in RSV-A G gene and another with a 60-nt duplication (BA genotype) in RSV-B at a similar region [20]. The observed RSV genetic diversity has raised a question about whether it is necessary for an RSV vaccine to include several different strains to be effective. Most current RSV vaccine developments are based on an RSV A2 laboratory strain, which is a chimeric strain that belongs to subtype A [21]. While these treatments hold promise, there is the possibility of viral strains developing escape mutations. For example, palivizumab-resistant strains have been isolated from both RSV rodent models and human [17,22]. Several lines of evidence also suggest antigenic variation may play a role in the ability of RSV to escape immune response and established infections [23]. While highly conserved T cell epitopes in RSV vaccine may not provide complete protection against infection when cross-protective antibody responses are lacking, highly conserved T cell epitopes in the vaccine may still reduce the severity of the illness and limit the spread of the virus. However, amino-acid variation at the T cell epitope level and the potential emergence of novel T cell epitopes of recent RSV circulating strains have been reported [24], and further studies are needed to illustrate the effects of amino acid variations on T cell recognition. Hence, characterizing T cell epitope profiles across different strains is very important to understand RSV evolution and can be important for RSV vaccine development.

In this study, we utilize immunoinformatic approaches that are implemented in the iVAX toolkit [25] to predict T cell epitopes in RSV across different strains with a focus on the two major surface proteins F and G. With the analysis of a comprehensive dataset, we evaluate the lineage-specific T cell epitope profile of RSV. We also create sequence-based T cell epitope landscapes based on epitope content comparison across different strains and further correlate RSV T cell immunity change with virus evolution. The proportion of cross-conserved T cell epitope content between vaccine candidate strains that developed earlier and RSV circulating strains with different isolation years and locations was also calculated. These analyses may aid in understanding RSV T cell immunity across different strains and contribute to current vaccine design efforts.

## Results

### Distribution of T cell epitopes in RSV surface proteins

We evaluated the T cell immunogenic potential across RSV surface proteins by scanning 9 residue regions to predict the binding probability to MHC class I and class II molecules (Fig 1). The epitope density of RSV surface proteins was evaluated using a normalized epitope density score, which is computed by summing up the predicted peptide-MHC binding score across the protein and normalizing it with the protein length. The score for randomly generated

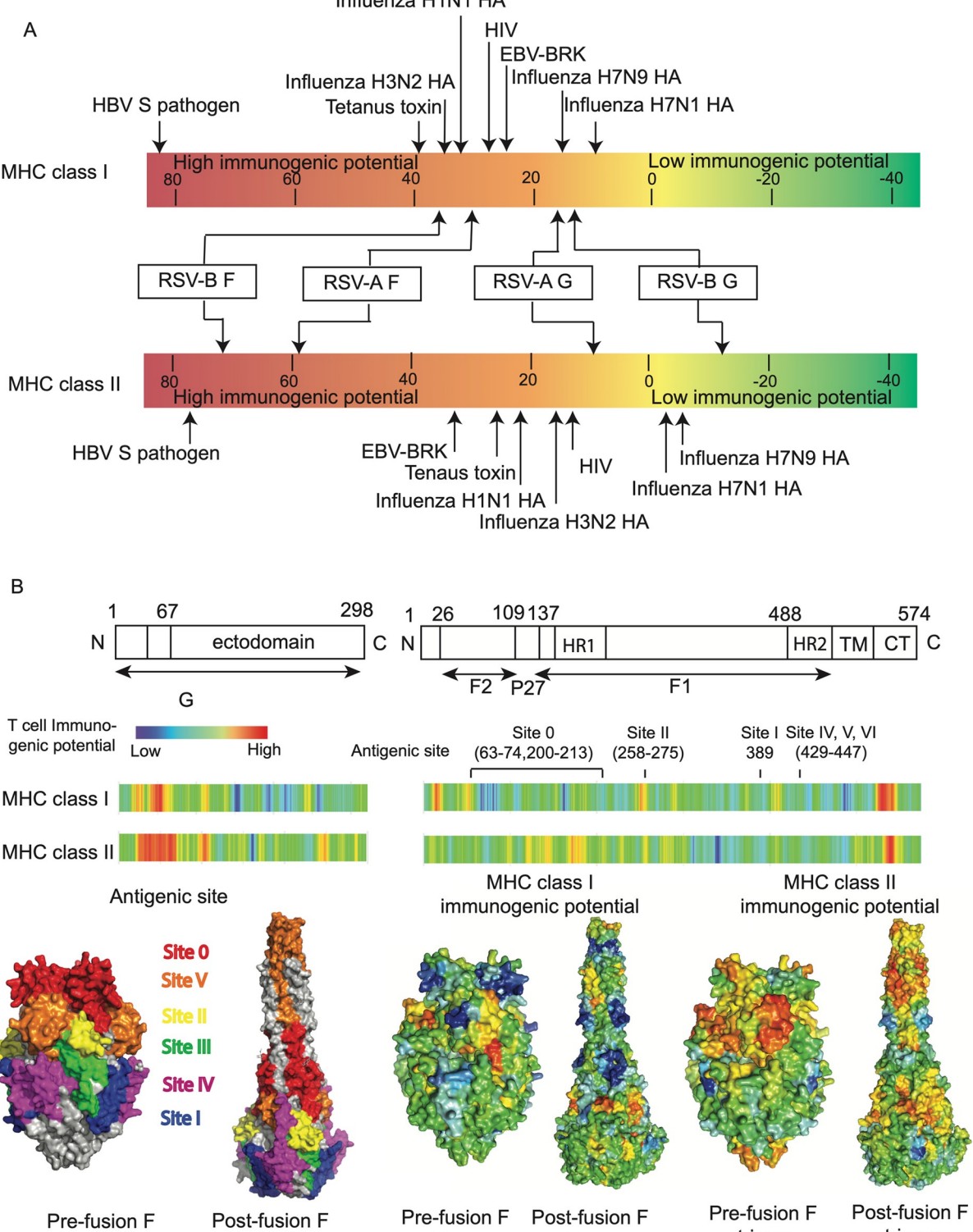

**Fig 1. T cell immunogenic potential for RSV surface proteins based on MHC binding prediction.** (A) T cell immunogenic potential of RSV major surface proteins. T cell epitope density scores for RSV major surface proteins and other pathogen proteins are labeled on a scale bar. Low-scoring proteins are known to engender little to no immunogenicity while higher-scoring proteins are known immunogens. Proteins scoring above +20 on this scale are considered to have significant immunogenic potential. (B) Distribution of RSV T cell immunogenic potential across F and G protein in RSV reference strain A2 and RSV F protein main antigenic sites that are determined in previous studies

[26]. Prefusion or post-fusion F protein surface was colored by the antigenic sites and relative immunogenetic potential at each location. Analyses are based on the RSV-A reference sequence.

proteins is set to zero and vaccine antigens generally score above +20 on this scale [25]. F protein has an epitope density score greater than +20 for both the class I and class II immunogenicity scale analysis, indicating significant immunogenic potential [25]. This contrasts with lower G protein class I and class II epitope density protein scores for both RSV subtypes. The class I epitope density score of G protein was greater than +10 in both subtypes but the class II density was lower than random expectation in the analysis of RSV-B (Fig 1A). This result suggests that RSV surface proteins are likely to have the potential to stimulate T cells that are required for protective immunity. We then investigated the distribution of T cell immunogenicity across the proteins and found that there are regions with relatively high T cell immunogenic potential (Fig 1B). The distribution of T cell immunogenicity of F protein was mapped onto its protein structure and overlap between protein sequence regions with high T cell immunity potential and the antibody neutralizing targets was observed at antigenic site Φ and site II.

## Lineage specific T cell epitope profiles

We then extended T cell epitope predictions from RSV representative strains to multiple wild-circulating strains. The distribution and diversity of T cell epitopes across different strains are illustrated in heatmaps with the corresponding time-scaled phylogenies (S1 and S2 Figs). Both F and G proteins contain epitopes that were conserved across all RSV strains in almost 100% of sampled isolates, suggesting that they could serve as high-quality T cell epitope candidates for vaccine design. In contrast, some epitopes were mutated in selected strains, and those epitopes that only occurred in certain clades within the phylogeny could be interpreted as clade-specific "fingerprints".

The G gene duplication events in RSV, which are unique gene signatures, can either shift the position of epitopes (locations are different but the amino acids of epitopes are identical to the G protein isolates without duplication) or cause the emergence of novel epitopes. Two novel class I epitopes, (no. 31 and no. 40 in S2A Fig), were found in RSV-A strains that contain G gene duplication. In addition, an emergent class II epitope (no. 25 in S2A Fig) was identified in RSV-A sequences that contain G gene duplication, which was a shift from an epitope (no. 24) that has been observed in other strains. From RSV-B strains that contain the G gene duplication event, we also observed multiple lineage specific class I T cell epitopes, which are caused by a 2-amino acid (aa) deletion (aa157 and aa158) in these strains instead of directly due to the 60-nt duplication event. RSV-B G proteins that have the duplication event contain multiple novel epitopes (no. 22, 23, 26, 28, 30, 37) but do not contain several epitopes (no. 24, 25, 27, 29, 31, 38) that are identified in other strains (S2B Fig).

To further determine whether the computationally predicted T cell epitopes with high MHC binding potential are immunogenic, we utilized the JanusMatrix [27] algorithm to identify the T cell epitopes that are likely to be cross-conserved with human peptides and thereby tolerated by the immune system. Based on this analysis, 6.45% of putative class I epitopes and 1.12% of putative class II epitopes of RSV major surface proteins are cross-conserved with human proteome-derived epitopes at T cell receptor (TCR)-facing residues. As these peptides have similar HLA binding preferences that are contained in human proteins (S3 Fig), they were therefore assumed not to be immunogenic. After excluding the high-JanusMatrix score epitopes identified above, we were able to identify T cell epitopes that were conserved in more than 60% of currently circulating RSV strains. We searched the IEDB epitope database to determine if these epitopes were related to experimentally validated RSV T cell epitopes or HLA ligands (Table A

**Table 1. Experimentally validated conserved MHC class I epitopes peptides in RSV major surface proteins[a].**

| Subgroup | Protein | Epitope address | Epitope sequence [b] | Binding HLAs [c] | Conservation [d] | Number of human matches [e] | Epitope id in IEDB |
|---|---|---|---|---|---|---|---|
| RSV-A & RSV-B | F | 45–53 | **LSALRTGWY** | A0101 | 99.55%(A) & 74.24%(B) | 1 | 158982 |
| | | 140–148 | **FLLGVGSAI** | A0201 | 99.59%(A) & 97.98%(B) | 0 | 156869 |
| | | 250–258 | **YMLTNSELL** | A0201, A2402 | 99.59%(A) & 99.33%(B) | 0 | 156979 |
| | | 272–280 | **KLMSSNVQI** | A0201 | 66.64%(A) & 96.08%(B) | 3 | 156902 |
| | | 273–281 | **LMSSNVQIV** | A0201 | 66.56%(A) & 96.08%(B) | 1 | 156915 |
| | | 449–457 | **TVSVGNTLY** | A0101 | 99.75%(A) & 99.33%(B) | 0 | 97017 |
| RSV-A | F | 10–18 | AITTILAAV | A0201 | 84.69% | 3 | 156844 |
| | | 111–119 | LPRFMNYTL | B0702 | 91.18% | 0 | 158975 |
| | | 170–178 | ALLSTNKAV | A0201 | 99.67% | 2 | 156847 |
| | | 383–391 | NIDIFNPKY | A0101 | 95.86% | 0 | 159045 |
| | G | 25–33 | FISSCLYKL | A0201 | 99.26% | 0 | 158759 |
| | | 61–69 | FIASANHKV | A0201 | 82.08% | 0 | 158751 |
| RSV-B | F | 525–533 | IMITAIIIV | A0201 | 89.25% | 0 | 156892 |
| | | 540–548 | SLIAIGLLL | A0201 | 97.65% | 5 | 156960 |
| | G | 25–33 | VISSCLYKL | A0201 | 90.91% | 0 | 158759 |
| | | 61–69 | FIISANHKV | A0201 | 99.02% | 0 | 158751 |

a. This table contains putative MHC class I epitopes that have already been experimentally validated in publications. Only putative class I epitopes that have positive results in MHC class I ligand assays with the same computationally predicted binding HLAs are shown in the table.

b. Epitope sequences that are conserved in both RSV-A and RSV-B are in bold.

c. HLAs that have the top 1% binder scores in EpiMatrix for epitope sequence.

d. The conservation is evaluated by the presence of epitope peptides across all RSV-A or RSV-B sequences that are publicly available (only epitope sequences with at least 60% conservation are shown in the table).

e. Count of human peptides found in the search database. JanusMatrix was used to search human peptides that are predicted to bind to the same allele as the RSV epitope and share TCR-facing contacts with the RSV epitope.

in S1 Text). The conserved RSV T cell epitope sequences that may be important for future vaccine development are shown in Tables 1 and 2 (Table B and C in S1 Text).

## Predicted RSV T cell epitope landscapes

To investigate the evolution of RSV on T cell immunity profiles, we use a multidimensional scaling (MDS) approach to visualize the T cell immunity profile of multiple RSV strains on a landscape. We performed a T cell epitope content pairwise comparison between RSV strains using *in silico* predicted peptide-HLA allele binding affinity. The pairwise T cell epitope distances were then calculated using the algorithm reported in this study (Eq 1). We then applied a multidimensional scaling (MDS) approach using these estimated pair-wise T epitope distances to map RSV strains to a landscape to characterize their T-cell immunity profile. We found both class I and class II T cell immunity profiles of F and G proteins of different RSV strains were clustered into groups on this T cell epitope landscapes (S4 Fig). Combining the class I and class II T-cell epitope binding profiles, RSV-A major surface protein isolates can be divided into three clusters and RSV-B major surface protein isolates can be divided into two clusters (Figs 2 and S5). We observe that the G gene sequence isolates that contain 72-nt (RSV-A) or 60-nt (RSV-B) duplications clustered together with other sequences instead of forming isolated groups. To further investigate the T cell epitope diversity, we correlated this clustering pattern with the phylogenetic histories (Fig 2B). The phylogenetic tree topologies of the RSV-A F gene and G gene are similar. The F gene cluster 1 is paraphyletic, while clusters 2 and 3 are monophyletic. Cluster 1 is the closest to the ancestral sequence and mapping this

**Table 2. Experimentally validated conserved MHC class II epitopes peptides in RSV major surface proteins[a].**

| Subtype | Protein | Epitope address | Epitope sequence [b] | Conservation [c] | Number of human matches [d] | Epitope id in IEDB |
|---|---|---|---|---|---|---|
| RSV-A | F | 29–44 | TEE**FYQSTCSAVS**KGY | 98.53% | 3 | 956680 |
| | | 50–70 | TGW**YTSVITIELSNIK**ENKCN | 97.75% | 1 | 153700 |
| | | 167–192 | IKSALLSTNKAVVSLSNGVSVLTSKV | 93.14% | 4 | 545502 |
| | | 218–234 | ETVIEFQQKNNRLLEIT | 98.86% | 3 | 1087566 |
| | | 247–268 | VSTYMLTNSELLSLINDMPITN | 98.98% | 8 | 99471 |
| | | 288–310 | IMSIIKEEVLAYVVQLPLYGVID | 98.57% | 5 | 99334 |
| | | 399–418 | KTDVSSSV**ITSLGAIVS**CYG | 99.14% | 0 | 545603 |
| | | 453–470 | GNTLYYVNKQEGKSLYVK | 98.37% | 1 | 99691 |
| | | 492–510 | ISQVNEKI**NQSLAFIR**KSD | 80.32% | 1 | 153713 |
| | | 543–560 | AVG**LLLYCKARSTPV**TLS | 79.26% | 6 | 153641 |
| | G | 19–43 | TLNHLLFISSCLYKLNLKSIAQITL | 93.13% | 8 | 1087567 |
| RSV-B | F | 29–44 | TEE**FYQSTCSAVS**RGY | 99.78% | 3 | 956680 |
| | | 50–70 | TGW**YTSVITIELSNIK**ETKCN | 93.95% | 1 | 153700 |
| | | 192–218 | VLD**LKNYINNQLLPIVNQQSCRISNIE** | 83.43% | 4 | 153636 |
| | | 247–268 | LSTYMLTNSELLSLINDMPITN | 98.54% | 8 | 99471 |
| | | 399–418 | KTDISSSV**ITSLGAIVS**CYG | 98.88% | 0 | 545603 |
| | | 453–470 | GNTLYYVNKLEGKNLYVK | 98.77% | 0 | 99691 |
| | | 492–510 | ISQVNEKI**NQSLAFIR**RSD | 97.42% | 1 | 153713 |
| | | 543–560 | AIG**LLLYCKAKNTPV**TLS | 94.96% | 4 | 153641 |
| | G | 51–74 | STSLIIAAIIFIISANHKVTLTTV | 94.66% | 8 | 158751 |

a. This table contains putative MHC class II epitopes that share the identical binding groove sequence, which represent the nine-mer frames with the greatest potential to bind class II HLA (epitope sequences with underlines), with the RSV class II epitopes that have already been experimentally validated in publications. Only the putative class II epitopes that have positive results in MHC class II ligand assays with the same computationally predicted binding HLAs are shown in the table.

b. Underlined sequences represent the nine-mer frames with the greatest potential to bind class II HLA. Epitope sequences that are in bold indicate sequences are predicted to bind class II HLA and are conserved in both RSV-A and RSV-B.

c. Conservation is evaluated by the presence of epitope peptides across all RSV-A or RSV-B sequences that are publicly available (Only epitope sequences with at least 60% conservation are shown in the table).

d. Count of human peptides found in the search database. JanusMatrix was used to search human peptides that are predicted to bind to the same allele as the RSV epitope and share TCR-facing contacts with the RSV epitope.

group onto the phylogeny shows that this cluster has a basal relationship with clusters 2 and 3 indicating that the phylogenetic divergence occurred prior to epitope drift. The RSV-B F and G gene genealogies are very different. In particular, the RSV-B F gene topologies is indicative of strong immune selection, similar to observed human influenza A virus or within host HIV phylogenies [28]. In contrast, the RSV-B G gene phylogeny shows the co-circulation of multiple lineages, though this could reflect the sequencing bias of G genes (Fig 2B). We then calculated the T-cell epitope immune distance of each strain from a reconstructed ancestral sequence (Fig 2C). These distances were then plotted against the year of isolation and colored according to the cluster identified in Fig 2A. RSV-A shows that multiple predicted immune phenotypes co-circulate and persist for long periods (>2 decades). Analysis of RSV-B shows a turnover of the predicted immune phenotypes with short periods of co-circulation (<5 years) for F and G protein T cell epitopes. The limited periods of co-circulation are again consistent with phenotype patterns observed for viruses under strong immune selection (e.g H3N2 influenza A virus) [29,30]. In contrast, genetic distances from the reconstructed ancestral sequence plotted against year of isolation show patterns typical of gradual genetic drift, except in the G

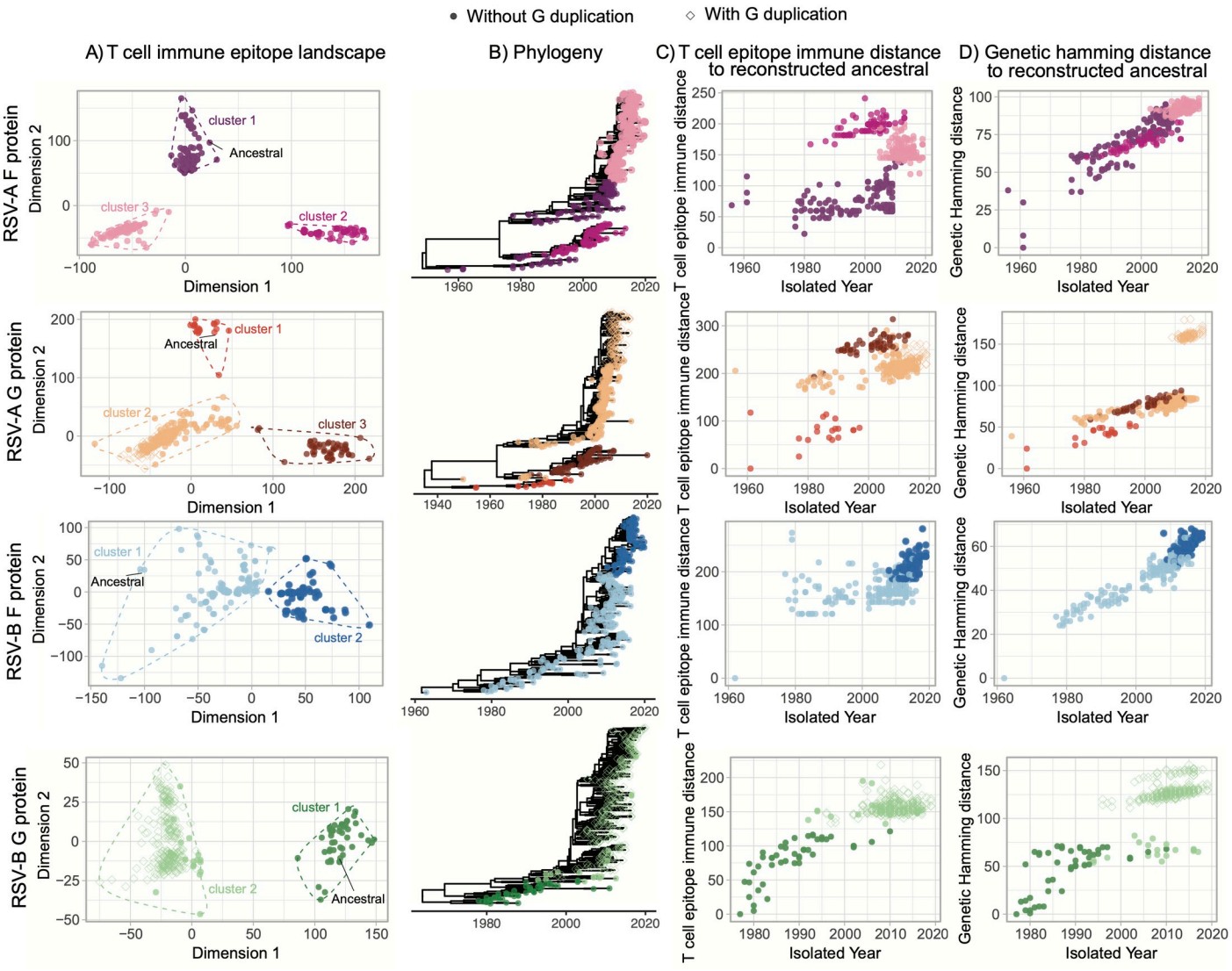

**Fig 2. Predicted T cell epitope landscapes and genetic evolution of RSV surface proteins.** Filled circles indicate RSV F protein isolates or G protein isolates without duplication. Diamonds indicate G protein isolates with gene duplication. (A) Epitope landscapes of RSV major surface proteins are built with MHC class I and class II epitope content comparison across different strains. T cell immunity clusters are determined with *k-means* method and are used to color the sequenced isolates in the following panels. (B) The corresponding time-scaled phylogenies are reconstructed with the Maximum Likelihood (ML) approach. (C) T cell epitope immune distance and (D) genetic hamming distance from the estimated TMRCA are plotted against the isolated time of each sequence.

gene where a 72-nt and 60-nt insertion is present (Fig 2D). Taken together, these results suggest that genetic and predicted T-cell epitope immune diversity are different and may be an important factor to consider when evaluating RSV vaccine efficacy.

There are multiple methods available to predict T cell epitopes [31], which may result in different reconstructed landscapes if there is a systematic bias in the prediction method. We used the NetMHCpan method [32] to predict T cell epitopes and perform the same landscape reconstruction using MHC class I binding predictions for RSV-A F protein. Our analysis showed a consistent clustered pattern of RSV T epitope profile on the landscape regardless of T cell epitope prediction method (S7 Fig).

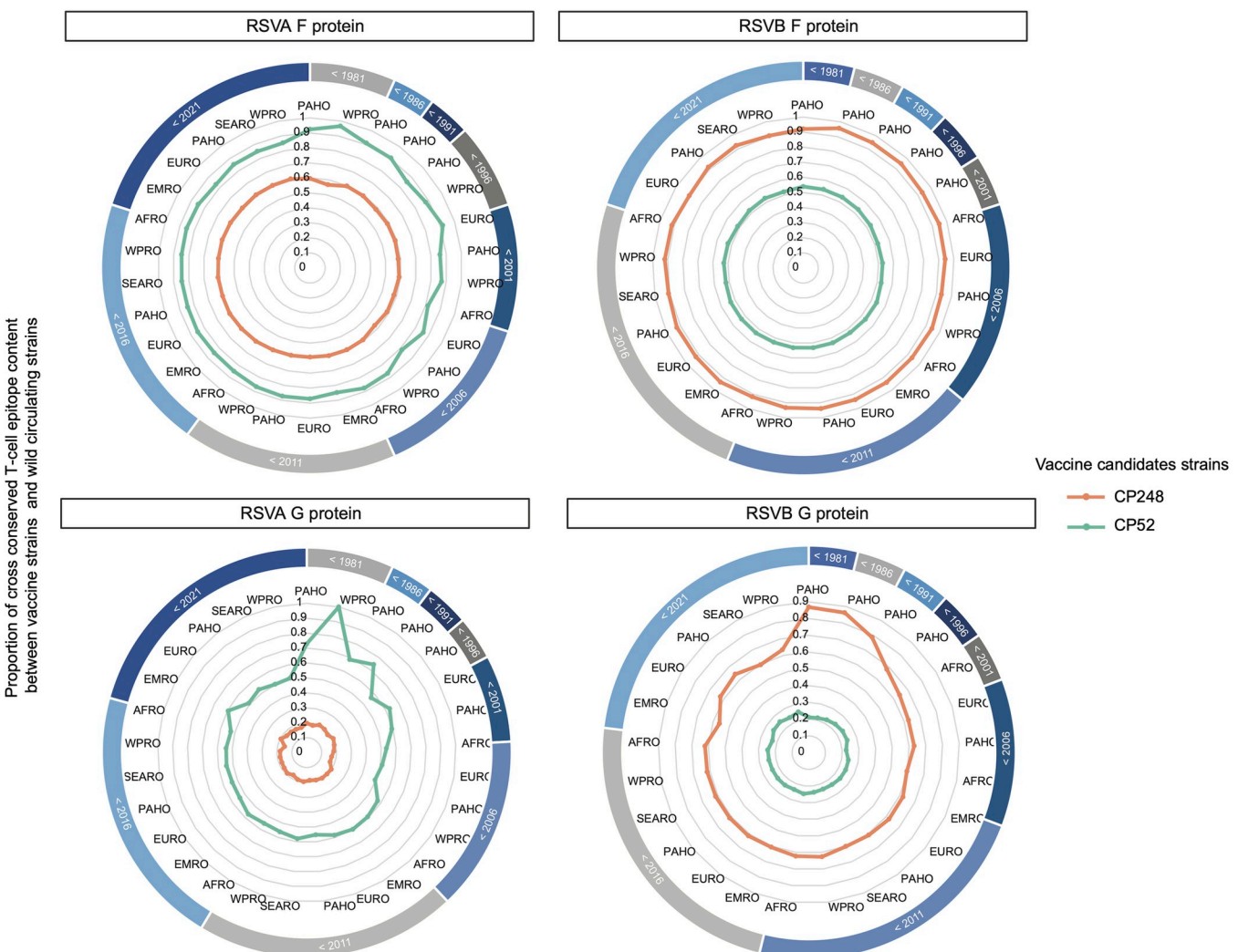

**Fig 3. Evaluation of previously used RSV vaccine candidate strains with T cell epitope content of circulating strains.** RSV-A and RSV-B major surface protein sequences are subsampled and then grouped by isolation year and 6 isolated WHO regions. African Region (AFRO), Region of the Americas (PAHO), South-East Asia Region (SEARO), European Region (EURO), Eastern Mediterranean Region (EMRO) and Western Pacific Region (WPRO). The proportion of cross-conserved T cell epitope content between live attenuated strains (CP248 or CP52) and wild circulating strains are displayed as radar plots.

## Assessment of vaccine candidate strains with T cell epitope content

T cell epitopes that are similar between vaccine strains and wild strains (cross-conserved T epitopes, which are defined as epitopes that share identical T cell receptor-facing residues and are restricted by the same alleles [33]) may be responsible for the T cell immune protection of the vaccine. To quantitatively evaluate whether it might be necessary to include multiple RSV strains to prepare an effective vaccine, two live attenuated RSV strains that are previously considered as vaccine candidates, CP248, a recombinant virus that belongs to subtype A, and CP52, which is a recombinant RSV-B strain, were included in our analysis and we evaluated their T cell epitope conservation with different RSV wild-type strains. We calculated the average proportion of cross-conserved T cell epitope content between the selected vaccine strains and wild-circulating strains from different isolation years and WHO regional groups (Figs 3 and S8). Different proportions of cross-conserved T cell epitope content against isolates from two different subtypes, A and B, were observed in both the F and G protein analyses. In the comparison of the vaccine strains and wild

strains belonging to the same subtype, the proportion of cross-conserved T cell epitope in RSV F protein is relatively stable in different groups, all are higher than 78% for RSV-A and higher than 85% for RSV-B. In contrast, changes in the proportion of cross-conserved T cell epitopes were detected among groups within the same RSV subtype, especially in different temporal groups in the G protein analysis (Fig 3). Vaccine strain CP248 appears to have a relatively higher proportion of cross-conserved T cell epitopes within G protein when compared to the RSV-A strains that were isolated before 1991 (> 70%) and a relatively lower degree of conservation against recently isolated strains. A similar decrease in T cell epitope conservation with time was identified for vaccine strain CP52 among circulating RSV-B strains.

## Discussion

Although both CD4 and CD8 T cells contribute to protection against RSV-induced disease following primary infection [16,34], T cell epitopes have received limited attention in the RSV research effort. We demonstrate RSV surface proteins appear to have significant potential to drive T cell immunity using a computational approach, based on their T cell epitope density scores as determined by MHC molecular binding prediction. The relatively high putative T-cell epitope density might make F protein a good target for RSV vaccine. In addition to the analysis of T cell epitope density and distribution in RSV major surface proteins, we also demonstrated lineage-specific variations in T cell epitope content. Even though RSV F protein is believed to be well conserved, epitope mutations are observed across different lineages within the F protein, suggesting that studying the lineage-specific T cell epitopes in RSV can provide insight into the impact of immune selection on viral diversity and persistence. In contrast to the conserved F protein, RSV G protein is reported to be highly variable, we still observed potential conserved T cell epitopes across different strains, which suggests a great interest of G protein conserved domain as potential vaccine targets on T cell immune protection. While experimental validation is needed, this analysis highlights the importance of understanding population-level epitope conservation as it may provide important insight into the development of T cell epitope-driven vaccines against RSV infection.

A major focus of our work is the development of a sequence-based method to map the evolution of T cell immunity across different stains. Following a previously pivotal work that used MDS method to map the evolutionary adaptation of influenza A virus-induced by CD8 T cell using the presence and absence of MHC class I epitopes [35], we constructed RSV T cell immunity landscapes using immune distances that were generated by T cell epitope cross-conservation analyses, which allows for easy visualization and intuitive understanding of the potential for T cell immunity relationships among different strains. When comparing across strains, we found that the T cell epitope content of RSV surface proteins from different strains can be clustered, as has been observed for the antigenic relationship reported in other pathogens [36,37]. Our results also demonstrate the correspondence between RSV T cell immunity clusters and their corresponding phylogeny, with sequences in the same clade generally belonging to the same T cell immune cluster. Importantly, we also observe different patterns of T-cell epitope evolution of RSV wild strains compared with their genetic evolution. We find the RSV strains with G gene duplications can still cluster with previous RSV isolate on the T cell epitope MDS space. Our results highlight the importance of characterizing T cell epitope changes in RSV.

We identified highly conserved RSV T cell epitopes in this study, some of which have already been experimentally validated and published in the IEDB database. However, we also identified several other conserved T cell epitopes that have not been previously described. These may be valuable for vaccine design, although experimental validation will be needed. Furthermore, the homology of selected RSV epitopes to human peptides suggests that some predicted RSV T cell

epitopes might be tolerated by the human immune system, or could induce a harmful cross-reactive immune response against human proteins when administered with an adjuvant [27]. Certain aspects of immunity to RSV were not addressed by this study. For example, neutralizing antibody responses are currently considered to be the most important correlate of immunity. While neutralizing antibodies would not directly be elicited by a T cell epitope-driven vaccine, helper (CD4) T cell epitopes are required to generate high affinity, high specificity antibodies. We also note that we have limited our focus on the two major RSV surface proteins in our current analysis, but other RSV proteins like N, M, or M2-2 proteins might also contribute to vaccine efficacy [38].

An effective vaccine against variable viruses should contain T cell epitopes that are highly conserved among circulating strains [39]. Vaccine efficacy can be diminished if T cell epitopes in a vaccine strain do not match when new strains of pathogens emerge. In this study, we used an immunoinformatic-based approach to estimate cross-conserved T cell epitope contents between two live attenuated vaccine candidate strains and RSV circulating wild strains. We found that there was a low proportion of cross-conserved T cell epitope content with vaccine strains that belonged to different antigenic groups, which indicates the risk of using a single-subtype strain in RSV vaccines. In addition, we observed a lower proportion of cross-conserved G-protein T cell epitope content between vaccine strains and recent circulating strains in the same antigenic group, which suggests that including T cell epitopes from different strains in the same antigenic group might also be important for RSV vaccine development. Although we did not observe a significant change in cross-conserved T cell epitope content in F protein, we cannot rule out the possibility that variation of F protein in the future could render a single-strain-based vaccine less effective. Our current analysis is based on reduced datasets due to the heavy computational capacity required to perform epitope content comparison. We constructed these representative datasets by randomly subsampling the complete datasets according to geographical regions and isolated years. Our findings may reflect the T cell epitope diversity of publicly available RSV strains, however, additional RSV surveillance efforts may be required to get a full picture of the T cell epitope variability of RSV.

Our current study is limited by lacking experimental validation of T cell epitope prediction. We focus on computationally predicted MHC binding to identify T cell epitopes. Although the strength of MHC binding is the key parameter that determines a peptide's immunogenicity, but not sufficient for a module to be immunogenic [40]. Other aspects associated with pathogen-induced T cell immune response, such as appropriate antigen-processing [41] and T cell receptor recognition [40] are not considered in this study, which might cause bias in the computational-based T cell epitope landscapes. However, the observed clustered pattern of RSV surface proteins on the T cell epitope landscape in this study reflects the diversity of T cell epitopes within different strains. This finding provides valuable insights into virus evolution in the aspects of T cell immunity and can contribute to the strain selection for vaccine design.

Overall, this study provides a focused analysis of T cell epitopes in RSV major surface proteins using computational tools. We performed a comprehensive T cell epitope prediction for RSV showing the immunological relationship of T cell epitopes in RSV surface proteins. This study demonstrates that T cell epitope evolution may differ from genetic variation and provides a framework for developing an integrated epitope-based RSV vaccine and evaluation methods that could be used to optimize vaccination strategies.

## Materials and methods

### Dataset

RSV GenBank records files were retrieved from NCBI's GenBank nucleotide database (https://www.ncbi.nlm.nih.gov/nucleotide/) using the search term "HRSVA" or "HRSVB" on June 22,

2020. F and G gene nucleotide sequences and metadata including country of isolation and collection date were extracted using customized python scripts. Genotype assignments were made with the program "LABEL", using a customized RSV module [42,43]. Countries of isolation were grouped into 6 WHO regions: African Region, Region of the Americas, South-East Asia Region, European Region, Eastern Mediterranean Region, and Western Pacific Region [44]. The following inclusion and exclusion criteria were applied: (i) each sequence needed to have a known isolated geographic location and isolated year, (ii) each sequence had to be at least 80% of the complete gene sequence in length, (iii) identical sequences with the same isolate country were removed, and (iv) vaccine derivative and recombinant sequences were removed. Using these criteria, comprehensive datasets of RSV F and G genes were defined (RSV-A F gene = 1010, RSV-B F gene = 894, RSV-A G gene = 1488, RSV-B G gene = 1120). Nucleotide sequences from each dataset were aligned using MAFFT.v7 [45] and were translated into amino acids using EMBOSS.v6.6.0 [46] for immunoinformatic analyses. In addition, two artificial sequences, CP248 and CP52 (cold passage live RSV strains that were previously evaluated as vaccine candidates, Accession No: U63644, AF0132551 respectively) were downloaded from the NCBI's GenBank nucleotide database [47].

## Phylogenetic inference

The nucleotide sequences of RSV major surface proteins were used to reconstruct the maximum-likelihood (ML) phylogeny of RSV using RAxML.v8 with GTR+GAMMA substitution model [48]. The best-scoring ML tree was automatically generated from five runs by RAxML. Time-scaled phylogenies were further reconstructed with the best scoring ML trees using the program "Timetree" [49]. The phylogenies are visualized in the R package "ggtree" [50].

## T cell epitope prediction

RSV major surface protein sequences were scored for binding potential against a globally representative panel of Human Leukocyte Antigen (HLA) class I and class II alleles using the EpiMatrix algorithm. This algorithm as well as the ClustiMer, JanusMatrix, and EpiCC algorithms discussed below are part of the iVAX toolkit developed by EpiVax, which is available for use under a license or through academic collaborations [25].

Evaluation of class I epitopes was made based on predictions for four HLA-A and two HLA-B supertype alleles: A*01:01, A*02:01, A*03:01, A*24:02, B*07:02, B*44:03. Class II epitopes were identified for nine HLA-DR supertype alleles: DRB1*01:01, DRB1*03:01, DRB1*04:01, DRB1*07:01, DRB1*08:01, DRB1*09:01, DRB1*11:01, DRB1*13:01, and DRB1*15:01. These are HLA allele supertypes (alleles sharing common binding preferences) that cover the genetic diversity of more than 95% of human populations globally [51,52]. EpiMatrix parsed 9-mer sequence frames (each one overlapping the previous one by eight amino acids) from the antigen sequence and assigned a score for each nine-mer/allele pair on a normalized Z distribution. Nine-mer sequences that had Z-scores of at least 1.64 are considered to be in the top 5% of any randomly generated set of 9-mer sequences and to have a high likelihood of binding to HLA molecules and being presented to T cells. Sequences that score above 2.32 on the Z-scale (top 1%) are extremely likely to bind to a particular HLA allele and to be immunogenic. For this analysis, HLA-class I restricted 9-mer sequences that had top 1% binder scores to at least one HLA class I supertype allele were considered to be putative class I epitopes [25].

To identify putative class II epitopes, we used an algorithm called ClustiMer [25] to screen EpiMatrix scoring results for the nine class II alleles. ClustiMer identifies contiguous regions of 15–30 amino acids that have a high density of MHC class II binding potential. Epitope

density within a cluster is reported as an EpiMatrix Cluster Score, where scores of 10 and above are likely to be recognized in the context of multiple class II alleles and to be high-quality class II epitopes.

### Identification of cross-conservation between putative RSV epitopes and human peptides

We also applied analysis of human homology to this study. After identifying putative T cell epitopes sequences in RSV major surface proteins, the JanusMatrix algorithm [27] was used to assess the potential cross-conservation of T cell epitopes with epitopes restricted by the same HLA alleles in the human proteome (Uniprot-sourced human proteins [53]). JanusMatrix scans input peptides and takes the 9-mer epitope regions that are identified in EpiMatrix to find the human peptides with a compatible HLA facing-agretope (i.e. the agretopes of both the input peptide and its human counterpart are predicted to bind to the same HLA allele) and the same TCR facing epitope to compute as a JanusMatrix Human Homology Score. As defined in retrospective studies, foreign class I epitopes that score greater than 2 and class II epitopes that score greater than 5 may be less immunogenic due to T cell tolerance [25].

### Protein-level T cell immunogenic potential evaluation

RSV reference sequences (RSV-A: NC_038235, RSV-B: NC_001781) were downloaded from the NCBI RefSeq database and were used to evaluate the protein-level immunogenic potential of RSV major surface proteins. The protein-level immunogenic potential as represented by the EpiMatrix-defined T cell epitope density score was computed by summing the top 5% binder scores across HLA alleles and normalizing for a 1000-amino acid protein length. Zero on this scale is set to indicate the average number of top 5% binders that would be observed in 10,000 random protein sequences with natural amino acid frequencies. Proteins scoring above +20 have been observed to have the significant immunogenic potential [54]. Fully human proteins generally score lower than zero on the EpiMatrix immunogenicity scale.

To investigate the distribution of T cell immunogenic potential across RSV protein sequence regions, we summed up the binding scores of HLA alleles for each nine-mer frame, to get a frame-specific immunogenic potential score and standardized this score to a relative scale. The relative immunogenic potential across protein structure was represented by a color scale and the visualization of F protein structure was built with PyMOL Molecular Graphics System, Version 2.0 (Schrödinger, LLC). Protein data bank (PDB) files 5UDE [55] and 3RRR [56] were used for the pre-fusion and post-fusion forms.

### Subsampling strategy

Considering the heavy computational load that would be required to evaluate all available RSV sequences and to correct the overrepresentation of recently sampled strains, the comparative analysis for T cell epitope content was conducted with datasets in which overrepresented groups were reduced. A maximum of five sequences of each isolation year from different WHO region groups were subsampled randomly from the original datasets (RSV-A F gene = 402, RSV-B F gene = 319, RSV-A G gene = 390, RSV-B G gene = 359).

### T cell epitope content comparison

The Epitope Content Comparison (EpiCC) algorithm, which is implemented in iVAX was used to compare T cell epitope content within each subsampled dataset by evaluating cross-conserved T cell epitopes (9-mer peptides with identical TCR-facing residues and are

predicted to binding to the same MHC allele) content between different virus strains [33]. We reasoned that epitopes with identical T cell receptor-facing residues (TCR$f$, position 4, 5, 6, 7, 8 for class I epitopes binding core and 2, 3, 5, 7, 8 for class II epitopes binding core), regardless of differences on their MHC-facing (*MHCf*) amino acids, which are also predicted to bind to the same MHC allele, are more likely to induce cross-reactive memory T cells (These epitopes are called cross-conserved T cell epitopes). To simplify the analysis, the binding of 9-mer epitopes within protein sequences are assumed to be mutually exclusive and uniform, which means the T cell immune response of the antigen protein can be represent by summing up all T cell epitopes within the protein sequence.

We use $u$ to represent 9-mer peptides with similar *MHCf* capable of binding the same MHC alleles but bearing different *TCRf* (non-cross conserved T cell epitopes) in two wild circulating strains ($w_1$ and $w_2$). Because the T cell immune response to virus is directly related to its T cell epitope content, the T cell immune distance (D) between two strains can be represented by the sum of binding probabilities of these unique 9-mer peptides for a set of HLA alleles (Eq 1). $p(u)_a$ is the predicted binding probability of unique 9-mer peptide $u$ against a single class I or class II allele $a$, that is a member of A, which represents a set of HLA alleles.

$$D(w_1, w_2) = \sum_{u \in (w_1, w_2)} \sum_{a \in A} p(u)_a \tag{1}$$

Since the calculation of T cell epitope immune distance relies on the predicted epitope binding affinity, we use another T cell epitope prediction tool to evaluate the T cell epitope immune distance generated by the EpiCC algorithm. We apply the Eq (1) to re-calculate T cell epitope immune distance with customized Python scripts (available at https://github.com/JianiC/RSV_Epitope/tree/master/NetMHCpan_reproduce) using MHC binding prediction results that are generated from publicly available T cell epitope prediction tool, netMHCpan EL 4.1 methods in the Immune Epitope Database (IEDB) [57]. Eigenvalues of each sequence that were calculated from the pairwise distance matrix with "RSpectra" package were used to statistically examine the correlation of the epitope distances that are computed from the two methods, and Pearson correlation test was used to test the correlation hypothesis.

The capacity for a vaccine to induce a T cell immune response that could be recalled by a wild circulating strain is related to the cross-conservation of the T cell epitopes between the vaccine strain ($v$) and the wild circulating strain ($w$). For each pair of 9-mer peptides $i$ (from strain $v$) and $j$ (from strain $w$) that are cross-conserved (i.e. bearing identical residues that face the TCR), the probability to recall cross-reactive T cell memory by those two 9-mer peptides via a single HLA allele $a$ can be represented by the joint estimation of the binding probability of these two 9-mer peptides ($p(i)_a * p(j)_a$).

Therefore, a T cell epitope similarity score (S) between two sequences can be represented by summing the probability to cross-reactive memory T cells by all paired 9-mer peptides that are cross-conserved between the vaccine strain ($v$) and wild circulating strain ($w$) against a set HLA alleles A (Eq 2.1).

$$S(v, w) = \sum_{i \in v, j \in w} \sum_{a \in A} (p(i)_a * p(j)_a) \tag{2.1}$$

We further normalized the T cell epitope similarity score between the vaccine strain and wild circulating strain by the maximum T cell epitope similarity score for the vaccine strain in

comparison with itself (Eq 2.2):

$$P(v, w) = \frac{\sum_{i \in v, j \in w} \sum_{a \in A} (p(i)_a * p(j)_a)}{\sum_{i \in v, j \in v} \sum_{a \in A} (p(i)_a * p(j)_a)} \quad (2.2)$$

## Dimension reduction

The equation to calculate T cell epitope immune distance was applied iteratively to the sub-sampled dataset and therefore the pairwise T cell epitope immune distances are structured into an $n \times n$ square-distance matrix. Given that each protein is described by a relative distance to the rest of n-1 proteins, the data must be dimensionally reduced to be graphed. Classic (metric) multidimensional scaling (MDS) can be used to preserve the distances between a set of observations in a way that allows the distances to be represented in a two-dimensional space. MDS was performed as previously described by Gower [58]. The MDS method first constructs an n-dimensional Euclidean space using the distance matrix in which all distances are conserved, and then principal component analysis is performed. MDS [59] were carried out using the *cmdscale* package in R [58]. *K-means* clustering was performed using the *kmeans* function in base R. Due to the lack of previous characterizations of RSV T cell immunity clusters, the number of T cell immunity groups was determined using the optimized within-cluster sum of square (wss) with Elbows method [60]. To evaluate whether applying *k-means* clustering to classify RSV strains on two-dimensional space can reflect their T epitope profile, we calculated the stress of MDS using the *smacof* package in R [61]. We also compare the performance of *k-means* clustering on MDS spaces with different numbers of dimensions (S6 Fig).

## Calculation of genetic hamming distance

Genetic hamming distance, which is defined as the number of bases by which two nucleotide sequences differ, was calculated by comparing the number of different bases between each sequence in the subsampled datasets. The reconstructed most recent common ancestor (TMRCA) sequences for each dataset (subsampled F and G protein sequences of subtype A and subtype B, respectively) were estimated using the program "Treetime" and were used as root in our analysis [49].

## Supporting information

**S1 Text. Supplementary Materials.** Table A: Number of computationally predicted conserved RSV T cell epitopes and experimentally identified RSV T cell epitopes. Table B: Conservation of experimentally validated conserved MHC class I epitopes peptides in RSV major surface proteins in subsampled dataset. Table C: Conservation of experimentally validated conserved MHC class II epitopes peptides in RSV major surface proteins in subsampled dataset. (DOCX)

**S1 File. Accession number to RSV sequence that are used in this study.** (CSV)

**S1 Fig. Distribution and diversity of T cell epitopes in RSV F protein.** The tree panel on the left is a time-scaled phylogeny build with RSV-A (A) or RSV-B (B) F gene nucleotide sequences using the ML approach. Determined genotypes are labeled on the right with black bars. Each color column on the right side represents the presence of an MHC class I or class II epitope. Only the epitopes that are present in more than 1% of sampled isolates are displayed. The column color indicates different numbers of epitope sequences at the same location. (TIF)

**S2 Fig. Distribution and diversity of T cell epitopes in RSV G protein.** The tree panel on the left is a time-scaled phylogeny build with RSV-A (A) or RSV-B (B) G gene nucleotide sequences using the ML approach. The clades that contain novel 72-nt or 60-nt duplication at the second hypervariable region of G gene were highlighted in red. Determined genotypes are labeled on the right with black bars. Each color column on the right side represents the presence of an MHC class I or class II epitope. Only the epitopes that are present in more than 1% of sampled isolates were displayed. The column color indicates different numbers of epitope sequences at the same location.
(TIF)

**S3 Fig. Distribution of JanusMatrix Human Homology score for putative RSV MHC class I and class II epitopes.** The cross-reactive potential of identified putative T cell epitopes and human host was represented with a JanusMatrix Human Homology score. 6.45% identified putative class I epitopes and 1.12% class II epitopes are cross-conserved on the TCR face with human peptides.
(TIF)

**S4 Fig. Predicted T cell epitope landscapes of RSV surface proteins.** RSV T cell epitope landscapes were built with sequenced-based MHC class I epitope binding prediction (left), MHC class II epitope binding prediction (middle) or combining class I and class II epitope binding prediction (right). Sequences are colored by the epitope cluster determined by epitope landscapes built with combining Class I and Class II epitope prediction
(TIF)

**S5 Fig. Total within sum of squares (*wss*) using *k-means* algorithm.** Totals within sum of squares in epitope topographies were calculated after clustering into k (from 1 to 10) groups with *k-means*. The optimal number of clusters is determined to be 3 in the analysis of RSV-A F and G proteins and is determined to be 2 in the analysis of RSV-B F and G proteins using the Elbow method.
(TIF)

**S6 Fig. Sensitivity analysis for MDS and *k-means* clustering.** (A) Stress evaluation under the different number of dimensions for RSV distance matrix. Stress less than 0.15 (red dash line) indicates an acceptable precise MDS solution. (B) Performance of *k-means* clustering under the different number of dimensions, the number of clusters is determined at 2-dimensional space. There is no cluster grouping difference at higher dimensional space (orange). Sum square between clusters /sum square of total differences (BSS/TSS) measures indicates the total variance in the data is explained well under higher dimensional space (green).
(TIF)

**S7 Fig. Validation of T cell epitope distance estimation using the IEDB analysis resource.** Validation is performed with MHC class I epitope binding prediction of RSV-A F protein. (A) Heatmaps for pairwise MHC class I epitope distance estimated in iVAX toolkits or calculated with custom python scripts using MHC class I molecule binding prediction that is implemented in IEDB. (B) Eigenvalues for each sequence are calculated from pairwise distance matrices using "RSpectra" package in R. The Pearson correlation test significantly supports a non-zero correlation between T cell epitope distance estimated with EpiCC and T cell epitope distance estimated with IEDB. (C) T cell epitope topographies are built with pairwise epitope distances estimated from EpiCC or IEDB. Both methods resulted in a similar cluster pattern for the CD8 T cell epitope profile of RSV-A F protein.
(TIF)

**S8 Fig. Evaluation of RSV vaccine candidate strains with class I and class II T cell epitope content in different WHO regions.** RSV-A and RSV-B major surface protein sequences were grouped by isolation year and 6 isolated WHO regions, African Region (AFRO), Region of the Americas (PAHO), South-East Asia Region (SEARO), European Region (EURO), Eastern Mediterranean Region (EMRO) and Western Pacific Region (WPRO). The proportion of cross-conserved T cell epitope content between vaccine strains (CP248 or CP52) and wild circulating strains in different isolation years and different WHO regions were represented. (TIF)

## Author Contributions

**Conceptualization:** Jiani Chen, Justin Bahl.

**Data curation:** Jiani Chen, Swan Tan.

**Formal analysis:** Jiani Chen, Swan Tan.

**Funding acquisition:** Justin Bahl.

**Methodology:** Jiani Chen, Leonard Moise, Anne S. De Groot.

**Project administration:** Justin Bahl.

**Resources:** Justin Bahl.

**Software:** Leonard Moise, Anne S. De Groot.

**Supervision:** Justin Bahl.

**Validation:** Vasanthi Avadhanula, Leonard Moise, Pedro A. Piedra, Anne S. De Groot.

**Visualization:** Jiani Chen.

**Writing – original draft:** Jiani Chen.

**Writing – review & editing:** Jiani Chen, Swan Tan, Vasanthi Avadhanula, Leonard Moise, Pedro A. Piedra, Anne S. De Groot, Justin Bahl.

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
