## [Decision Letter · Decision Letter 0]

19 Sep 2022

Dear Dr. Bahl,

Thank you very much for submitting your manuscript "Diversity and Evolution of Computationally Predicted T Cell Epitopes against Human Respiratory Syncytial Virus" for consideration at PLOS Computational Biology.

As with all papers reviewed by the journal, your manuscript was reviewed by members of the editorial board and by several independent reviewers. In light of the reviews (below this email), we would like to invite the resubmission of a significantly-revised version that takes into account the reviewers' comments.

Your manuscript has been reviewed by three knowledgeable reviewers. Although all three reviewers agree that your manuscript contains interesting information, they also had important major comments. I urge you to provide a precise and fully transparent point-by-point response to the comments by the reviewers, and revise your manuscript accordingly. As everyone is strongly time-constrained I may opt not to resend a revised manuscript to the reviewers, but rather make a decision based on your revised manuscript and response to the comments of the reviewers.

We cannot make any decision about publication until we have seen the revised manuscript and your response to the reviewers' comments. Your revised manuscript is also likely to be sent to reviewers for further evaluation.

Sincerely,

Michiel van Boven

Guest Editor

PLOS Computational Biology

Thomas Leitner

Section Editor

PLOS Computational Biology

Dear authors, your manuscript has been reviewed by three knowledgeable reviewers. Although all three reviewers agree that your manuscript contains interesting information, they also had important major comments. I urge you to provide a precise and fully transparent point-by-point response to the comments by the reviewers, and revise your manuscript accordingly. As everyone is strongly time-constrained I may opt not to resend a revised manuscript to the reviewers, but rather make a decision based on your revised manuscript and response to the comments of the reviewers.

Reviewer's Responses to Questions

**Comments to the Authors:**

Reviewer #1: The paper by Chen et al presents an analysis of T cell epitope evolution in RSV. Though an interesting study, the results remain much more descriptive than conclusive. Moreover, a number of issues are not clear and therefore it is rather difficult to re-produce the results presented in this paper. For example:

1) The paper makes use of a non-publicly available tool

2) Are only the epitopes presented in Table I and II are used to make MDS maps? How many epitopes were predicted by the method but not (yet) validated experimentally.

3) Vaccine section: what is the definition of being conserved? If an epitope still binds to its MHC, is it then a conserved epitope? T cell binding can still be totally lost.

More major points to re-consider:

1) At line 224 it is suggested that visualising T cell epitope evolution via MDS is a new approach. As the authors state it in the discussion, this is not the case.

2) JanusMatrix algorithm is used extensively in the paper, but the reader gets very little information on what this model does.

3) How were the alleles included in this study decided? Why are there only two HLA-B alleles and many more HLA-DRB alleles? Clearly allele choice might have a big impact on the results.

4) The analysis makes use of a lot of thresholds to identify a T cell epitope. How are these thresholds defined? Are the results sensitive to the thresholds.

5) Why are the Class I and II epitopes are combined in this analysis?

Minor points:

1) Both in Table I and II it would be nice to add what kind of experimental validation is found? in vitro binding, elution, T cell response?

2) Table II: are predicted and validated epitopes are the same or do they only have the same core?

3) Figure 2 caption: indicate what A, B, C and D panels are.

4) Which thresholds are used to make NetMHCpan analysis?

5) The sentence starting on line 78 does not sound correct to me.

6) Line 232: biding  binding

Reviewer #2: Chen et al investigate the diversity of RSV from the perspective of T-cell epitopes. These epitopes might be an interesting target for future vaccine development, both because they may provide broad immunity, but also because of the problem of vaccine-induced disease. The authors make use of of-the-shelf bioinformatic tools to find candidate CD4 and CD8 T cell epitopes for common HLA alleles. They filter out epitopes that are similar to human-protein derived peptides and compare the predicted epitopes with experimentally validated ones from IEDB. While I do think the approach can be useful and interesting, I have a couple of concerns that should be addressed. Most importantly, the method section has to be re-written and made much clearer.

Major comments:

The introduction was well written and is very informative, except for the last part of the fourth paragraph. I’m not sure what the relevance of cross-conserved epitopes is. I’m not sure what cross-conservation means. The final sentence does not follow naturally from this paragraph. Try making a better argument for why characterizing T-cell epitope profiles is important.

Identifying epitopes after a gene duplication event: On page 10 you mention that gene duplication events can “shift” the position of epitopes, or create novel epitopes. One could argue that a “shifted” epitope is the same epitope as the original. How is this counted when you compare strains in T-cell antigenic space? Are shifted epitopes identified, or counted as separate? How does this influence the clustering?

Clustering in low-dimensional space. As MDS is a non-linear dimension reduction method, clustering the data in the low-dimensional “MDS-space” could result in artifacts. For instance, points might cluster together because they are projected together, not because they are actually close in the original “epitope space”. This may not be an issue in this case, but I suggest checking this by first clustering the data, and then applying MDS. You might need a clustering method that works directly on the distance matrix (Leiden clustering perhaps?). Alternatively, you could increase the dimension of the MDS space, and use k-means on this higher-dimensional MDS space.

The definition of distance between strains. I don’t understand the logic behind equations 1.1 and 1.2. Why is this a good distance measure between strains? I also don’t understand the notation in 1.1. What is p(i)a? Are you multiplying p(i) with a? I thought that a is an HLA allele. You mention that you take z-scores of the probabilities p. At what point does this happen? Can distances be negative? It appears to me that something is wrong here. This has to be explained in a more rigorous manner.

The same holds for equations 2.1 and 2.2. Why switch to a different definition here (why don’t you use a single distance measure?). I don’t understand what you mean by “T-cell cross-conservation between two epitopes can be represented by a joint probability estimation and therefore …”. Why is 2.1 a product, while 1.1 was a sum?

Several sets of epitopes are mentioned: predicted epitopes (1%, 5%), then human peptides are filtered out, and then you list a set of experimentally confirmed epitopes. In the end, which set of epitopes is used for the MDS analysis and clustering?

Minor comments:

In the case of Influenza, some evidence exists that T-cell epitopes are subject to positive selection in humans. For instance Machkovech et al (J. Virol. 2015) compared the substitution rate in epitope sites with that of swine flu, and in Woolthuis et al (ref 32) the number of epitopes appears to decrease with time. It would be interesting to do a similar analysis for RSV. Have you tried this? Perhaps you can at least comment on this.

You give a list of experimentally verified epitopes. However, are there any epitopes in IEDB that are not present in the predicted dataset? If so, why are they not predicted?

Line 20: the B-cell response

Line 38: The B-cell epitope

Line 80: add some references? Or the names of some of these studies?

Line 105-108: I don’t understand this sentence. Please re-phrase.

Line 112: What does cross-conserved mean?

Line 125: do you mean “Isolation years”?

Line 133: Dependent on the final order of sections (Intoduction, Results, Discussion, Methods), at this point it is not clear how this score is generated. Perhaps give a one-sentence explanation and mention the +20 significance level.

Line 144: You mention that T-cell epitopes and Ab-epitopes overlap, but figure 1B does not show the location of Ab-epitopes. Would it be possible to indicate those in the 3d structures?

Fig S1: Is the “dominant” epitope the most common epitope or the epitope with the highest score? Somewhat confusing terminology.

Line 179: EpiMatrix has not been mentioned before.

Line 181: cross-conserved? Do you mean cross-reactive?

Table 1: Conservation is calculated using all available sequences. But did you account for sampling bias for more recent strains? If an epitope is present in a more recent strain, and many similar strains are also isolated, then the conservation is artificially inflated.

Line 203: perhaps say “human peptides” as these are (hopefully) not actual epitopes.

Line 238: Cluster 1 is paraphylectic, which is quite interesting, but might this also depend on the parameters/settings used for clustering. With alternative settings, cluster 1 could perhaps be split into 2 separate clusters that might become monophylectic. Can briefly you comment on this?

Line 281: Has EpiCC been introduced?

Figure 3: This is not a very clear representation of the data. Would a simple bar plot be better?

Line 363: “Possibility” instead of “probability”

Line 371: This paragraph is a bit weak. Can you say what these biases might be? And what are these valuable insights?

Line 423: “Each one overlapping the previous one by one amino acid”: surely you mean overlapping by 8 amino acids?

Line 480: What does this mean: “are assumed to be mutually exclusive and uniform”? Please explain this more clearly.

Line 496: where in the main text are you using this? Also, please explain more clearly what you are doing here (and why)

Line 521: what was the result of the goodness-of-fit analysis?

Line 528: How did you take these insertions into account in the genetic distance?

Reviewer #3: The paper is an interesting read and describes an elegant computational analyses that provides novel insights that improve understanding of immunity to RSV. I have some minor questions that may be addressed by the authors to broaden the discussion a bit:

1. The work takes a number of HLA types into account to find relevant epitopes. Considering the diversity of HLA alleles in a population of individuals that will be vaccinated, how would an individual's HLA typing impact the success of a vaccine that has been designed by this computational approach? In other words: does a selected peptide work out well for a large part of the whole population of vaccinees? Or should a wide selection be made from a large pool of different peptides? How do the authors envision such an approach towards vaccine design?

2. The authors stress the variations among RSV G proteins. However, the RSV G protein contains a central conserved domain that has been shown to provide T-cell immunogenicity. What would such conserved domain mean for predicting T-cell epitopes and immunity to G protein?

3a. Many epitopes are left out of the analysis due to similarity to human sequences, since this would generate tolerance. That is a fair point. However, these epitopes may also actively induce Treg cells that may control excessive T-cell responses causing unwanted inflammation. Can the authors exclude that such 'active control' as means of tolerance may be involved in the overall immune response to RSV? In other words, would neglecting such 'tolerant' epitopes pose a risk for excessive immune responses when regulatory T cells are not allowed to be activated sufficiently?

3b. Were epitopes in tables 1 and 2 indeed validated to be tolerated experimentally, where indicated? And was immunogenicity of these sequences checked and never observed?

Technical notes:

4. In figure 1B it might be helpful the reader to point antigenic sites ø and II in the 3D-structures of pre-F and post-F.

5. Figure 2A describes 'Tenaus toxin'. Is this Tetanus toxin?

**Have the authors made all data and (if applicable) computational code underlying the findings in their manuscript fully available?**

Reviewer #1: **No: **The authors are using a non-publicly available tool and their analysis is hard to reproduce because not all details are given. There is some code available in github but I wander if one can use it without having the predictions.

Reviewer #2: **No: **The authors use publicly available data, but did not share any of their scripts.

Reviewer #3: Yes

PLOS authors have the option to publish the peer review history of their article (what does this mean?). If published, this will include your full peer review and any attached files.

Reviewer #1: No

Reviewer #2: No

Reviewer #3: No
---

## [Editor Report · Decision Letter 1]

7 Dec 2022

Dear Dr. Bahl,

We are pleased to inform you that your manuscript 'Diversity and Evolution of Computationally Predicted T Cell Epitopes against Human Respiratory Syncytial Virus' has been provisionally accepted for publication in PLOS Computational Biology.

Best regards,

Michiel van Boven

Guest Editor

PLOS Computational Biology

Thomas Leitner

Section Editor

PLOS Computational Biology

---

## [Editor Report · Acceptance letter]

4 Jan 2023

PCOMPBIOL-D-22-01014R1 

Diversity and Evolution of Computationally Predicted T Cell Epitopes against Human Respiratory Syncytial Virus

Dear Dr Bahl,

I am pleased to inform you that your manuscript has been formally accepted for publication in PLOS Computational Biology. Your manuscript is now with our production department and you will be notified of the publication date in due course.

With kind regards,

Zsofi Zombor
